# Paramedic interactions with the packaging of medications and medical supplies: Poor package design has the potential to impact patient outcomes

Jiyon Lee[1☯], Rebecca E. Cash[2,3¤a‡], Remle P. Crowe[2,3¤b‡], Hyokyoung G. Hong[4‡], Ashish R. Panchal[2,3‡], Kami Silk[5], Marvin Helmker[6], Laura Bix[1☯]*

1 School of Packaging, Michigan State University, East Lansing, MI, United States of America, 2 Division of Epidemiology, College of Public Health, The Ohio State University, Columbus, OH, United States of America, 3 Department of Emergency Medicine, The Ohio State University Wexner Medical Center, Columbus, OH, United States of America, 4 Department of Statistics and Probability, Michigan State University, East Lansing, MI, United States of America, 5 Department of Communications, University of Delaware, Newark Delaware, Delaware, United States of America, 6 Community Paramedic Program, Lansing Community College (LCC), Lansing, MI, United States of America

☯ These authors contributed equally to this work.
¤a Current address: Department of Emergency Medicine, Massachusetts General Hospital, Boston, MA, United states of America
¤b Current address: ESO Solutions, Austin, TX, United states of America
‡ These authors also contributed equally to this work
* bixlaura@msu.edu

**Data Availability Statement:** All relevant data are within the manuscript and its supporting information files.

## Abstract

### Background

Settings where Emergency Medical Services (EMS) are provided to stabilize patients and transport them to locations better equipped to provide comprehensive care, "prehospital settings," are not frequently considered when designing packaged products. Packaging design is an understudied area, potentially impacting both healthcare provider behavior and patient outcomes. Our objectives were to: 1) describe difficulties associated with packaging in prehospital settings 2) investigate the coping strategies used by paramedics when difficulties occurred, and 3) assess the potential impacts these difficulties had on patient care.

### Methods

An online, cross-sectional survey was distributed via email using the National EMS Certification database maintained by the National Registry of Emergency Medical Technicians (NREMT) to a random sample of nationally-certified paramedics. Eligible respondents were aged 18 and older, employed as paramedics and had administered care in a prehospital setting within the previous 12 months. Survey items explored difficulties experienced and coping strategies used when difficulty was encountered identifying or opening medications and/ or medical supplies. Descriptive statistics and logistic regression were calculated to analyse responses for trends.

**Funding:** The Author(s) received no specific funding for this work. A portion of Laura Bix's salary is funded through a NIFA Hatch Act Grant provided by the USDA under project MICL02263.

**Competing interests:** The authors have declared that no competing interests exist.

## Results

Of the 12,000 emails sent, 1,912 participants responded (response rate = 16%). After removing respondents who had not administered care within the past 12 months and partial surveys, data from 1,702 respondents were analysed. Nearly 20% of all respondents reported that they had experienced difficulties identifying (21.1%) or opening (20.5%) medications and identifying (17.0%) or opening (23.4%) medical supplies within the past year. Between 1.2% (identifying a medication) and 3.0% (opening supplies) of those included in the analysis indicated that reported difficulties had negatively impacted patient care. Common coping strategies reported to deal with difficulty opening included partner assistance, tool use (scissors, pens, and knives), and the use of teeth, all potential pathways for the transmission of microbes, conceivably further impacting outcomes.

## Conclusion

More thoughtfully designed packaging for prehospital settings has the potential to benefit both EMS providers and the patients that they care for.

## Background

Engineering safety into the overall health care system to reduce errors and improve patient outcomes is an important paradigm that is being actively embraced by the designers of healthcare products and systems, caregivers and policy makers [1]. But research investigating the relationship of the design of packaged products and their impact on the behaviour of healthcare providers and patient care has not been widely conducted. This is likely because, historically, the regulators of medications and medical supplies have required the submission of objective evidence that manufacturing processes produce packaging that preserves, enables, and maintains the *product's* safety and efficacy throughout the expected shelf life. Although preserving, protecting and manufacturing products used for patient care is critically important, what has not been assessed is how the package design impacts the ability of the healthcare provider (or patient) to accomplish critical tasks (e.g. properly identifying, aseptically opening and transferring, and properly dosing and administering). Only recently have a host of standards and regulatory documents started to proliferate which require the objective evaluation of how packaging performs in the hands of users (both healthcare providers and patients) [2–6] (Table 1).

Even as standards and requirements for packaging change in an attempt to engineer safety into healthcare, little evidence exists describing the relationship of packaging design to provider behaviours, and ultimately, patient outcomes. This could be because the errors associated with packaging and labelling design are latent in nature, with the mistake occurring upstream from the presentation of the problem. Further, latent errors tend to be "systems related factors;" [16] that is, one factor among several that contribute to an event. Papers that review root-cause analysis of medical errors and adverse events suggest that similar packaging and labelling are fundamental reasons for medication errors [16–19], though other potential problems related to poor packaging design (e.g. difficulty opening or transferring cleanly) have yet to be thoroughly investigated.

And it isn't just packaging that hasn't been thoroughly researched, understood, and optimized; the healthcare environment itself, undoubtedly, impacts how care givers interact with

**Table 1. Objective evaluation and regulatory documents by standards.**

| Standards and Regulatory Bodies | Contents related to objective evaluation of how packaging performs in the hands of users. |
|---|---|
| **ISO 11607 Parts I** [3] | **6.1.1** The packaging system shall be designed to minimize the safety risks and health risks to the user and patient under the intended specified conditions of use. **6.1.8** If the packaging system to be opened at the point of use consists of more than one packaging layer, the sterile barrier system(s) shall have an indication to be recognized as such. **7.** Usability evaluation for aseptic presentation The usability evaluation for aseptic presentation shall include an assessment of *a) the ability to identify where to begin opening,* |
| **ISO 11156 General Requirements for Accessibility** [7] | 1 Scope . . ... a framework for design and evaluation of packages so that more people, including persons from different cultural and linguistic backgrounds, older persons and persons whose sensory, physical, and cognitive functions have been weakened or have allergies, can appropriately identify, handle and use the contents. |
| **ISO 17480 Ease of Opening** [8] | 1. Scope . . ... specifies requirements and recommendations for the accessible design for packaging with a focus on ease of opening. . .. the design aspects addressing openability including opening location, opening methods, as well as evaluation techniques, both instrumented and user-based. |
| **ISO/ DIS 19809 Information and Marking** [9] | 1. Scope . . .specifies considerations and methods for designing and presenting information and marking to make consumer packages accessible to people with the widest range of capabilities by taking account of their sensory and cognitive abilities. 4.1.1 Designing of information and marking of packaging shall take into considerations on diverse users and on diverse context of use. |
| **ISO 17351 Braille on Packaging for medicinal Products** [10] | 4.1 Principles of Braille legibility compliance The Braille text shall enable Braille readers to identify the medicinal product. Compliance with the Braille cell dot height limits is evidence of compliance with the text legibility requirement |
| **ISO CD 22015 Packaging- Accessible Design Handling and Manipulation** [11] | 1 Scope: . . . . . . in designing consumer packages, independent of material, to increase accessibility with regard to handling and manipulation. Handling and manipulation include human physical abilities like holding, lifting, carrying, pulling, pushing, sliding, grasping, twisting, tearing and any combination of those actions related to portability, opening, re-closing and taking out contents of packages as well as to storage and disposal. 6.3 User-based evaluation User-based evaluation enables packaging designs assessments and allows an understanding to develop of user's performance in handling and manipulation. User-based evaluation should be used in conjunction with other psychological methods, such as questionnaires and structured or unstructured interviews. The data generated by these user-based evaluations can provide insights for improved designs. |

*(Continued)*

**Table 1.** (Continued)

| Standards and Regulatory Bodies | Contents related to objective evaluation of how packaging performs in the hands of users. |
|---|---|
| **Applying Human Factors and Usability Engineering to Medical Devices: Guidance for Industry and Food and Drug Administration Staff** [12] | 2. Scope . . .focusing specifically on the user interface, where the user interface includes all points of interaction between the product and the user(s) including elements such as displays, controls, *packaging*, *product labels*, *instructions for use*, etc. While following these processes can be beneficial for optimizing user interfaces in other respects (e.g., maximizing ease of use, efficiency, and user satisfaction), FDA is primarily concerned that devices are safe and effective for the intended users, uses, and use environments. The goal is to ensure that the device user interface has been designed such that use errors that occur during use of the device that could cause harm or degrade medical treatment are either eliminated or reduced to the extent possible. |
| **U.S Food & Drug Administration-Code of Federal Regulations Title 21.820.30** [13] | (c) *Design input*. . . . the design requirements relating to a device are appropriate and address the intended use of the device, including the needs of the user and patient. . . (e) *Design review*. . . . formal documented reviews of the design results are planned and conducted at appropriate stages of the device's design development. . . .. participants at each design review include representatives of all functions concerned with the design stage being reviewed and an individual(s) who does not have direct responsibility for the design stage being reviewed, as well as any specialists needed. (f) *Design verification*. . . .Design verification shall confirm that the design output meets the design input requirements. . . (g) Design validation. . . .Design validation shall be performed under defined operating conditions on initial production units, lots, or batches, or their equivalents. Design validation shall ensure that devices conform to defined user needs and intended uses and shall include testing of production units under actual or simulated use conditions. Design validation shall include software validation and risk analysis, where appropriate. |
| **Medical Device Regulations (MDR)** [14] | Data gathered by the manufacturer's post-market surveillance system shall in particular be used: . . ... for the identification of options to improve the usability, performance and safety of the device; *Section 11.4* Devices delivered in a sterile state shall be designed, manufactured and packaged in accordance with appropriate procedures,. . . . . . until that packaging is *opened at the point of use*. It shall be ensured that the integrity of that packaging is clearly evident to the final user. |
| **Sterile Barrier Association (Ref. 201509 rev.01)** [15] | 4. **Open**: the seals of the sterile barrier system should be peeled slowly and evenly such that *a. the peeling is started on the specified side of the SBS b. the peeling direction is respected if specified . . . g. an opening is created large enough to remove the product without touching unsterile areas. (It may be necessary to fold the flaps backwards to create sufficiently large openings)* |

products to deliver patient care. Of particular interest to us was how packaging performs within the prehospital setting. Emergency Medical Services (EMS) are administered during prehospital care to stabilize patients and transport them to a location better equipped to

provide comprehensive care. Despite the extreme conditions that may be present (e.g., poor lighting, extreme heat or cold, noise, chaos, emergency vehicle movement, interloping friends and family), they are infrequently (if ever) considered by product and package designers.

As evidence of this, we offer data collected in healthcare transport environments, specifically intrahospital transports of critically ill patients, where the rates of adverse events have been noted to be as high as 70% [20–22]. Bergman et al. identified five broad categories of hazards contributing to transport-related adverse events and identified the most common to be errors associated with the category *tools and technology*, within this category a common problem was "products that were not designed in ways that consider the context/environment of care and support the needs [of heathcare providers] [23]."

Anecdotally, paramedics have been observed to employ flashlights to identify products, or use one hand, and scissors (or teeth) to open items required urgently. These coping strategies, used to deal with shortcomings of designs which don't appropriately consider the myriad of contexts where paramedics deliver care, have the potential to play a role in patient outcomes. For example, tools and/or inappropriate opening techniques may serve as an indirect mechanism for the transfer of microbes [17,24], a source of healthcare associated infections (HAIs). Although precise sources of microbes affiliated with HAIs are difficult to identify, a surveillance study investigating prehospital infection rates suggests that patients treated by advanced life support (ALS) providers prior to being admitted to the hospital have greater rates of HAIs than those transported by other means [25]. Supporting the idea that the prehospital environment is a fertile place for systemic improvement is the work of Orellana et al [24]. Among the factors that their research team identified as significant risk factors associated with Methicillin-resistant *Staphylococcus aureus* (MRSA) detection on EMS personnel were infrequent hand hygiene after glove use and low frequency of hand washing. Researchers found a high prevalence of MRSA among Ohio EMS personnel, noting this was both an occupational hazard and a patient safety concern [24].

It is unclear whether conditions present in prehospital environments affect the use of packaged products in ways that provide a source of risk for patients. In light of this, we evaluate the notion that prehospital work creates unique challenges for paramedics as they use packaged products to deliver care. In doing so, our objectives were to:

1. describe difficulties associated with packaging design in prehospital settings;

2. investigate the coping strategies used by paramedics when difficulties occurred; and

3. assess the potential impacts these difficulties had on patient care.

## Methods

The study was conducted in accordance with methods approved by the Institutional Review Board at the American Institutes for Research (AIR (#EX00411)), the review board affiliated with the National Registry of Emergency Medical Technicians (NREMT). The transfer of de-identified data to MSU for analysis was determined to not meet the definition of human subjects as defined by the US Department of Health and Human Services) by the MSU Institutional Review Board Manager (application #x16-1412e).

### Study design, population & setting

We conducted a cross-sectional survey of nationally certified paramedics included in the National EMS Certification database. Maintained by the National Registry of Emergency Medical Technicians (NREMT), the National EMS Certification database is the largest database of

EMS professionals in the United States, with 406,939 EMS professionals entered as of 2018 [26]. Power calculations were performed to determine the number of respondents needed to make estimates with 95% confidence. The calculated sample size was inflated assuming a conservative 10% response rate, leading to a simple, random sample of 12,000 paramedics drawn from the aforementioned database. The survey was limited to paramedics because they can provide a higher level of patient care and, therefore, were more likely to interact with multiple packaging types in the course of care delivery.

## Survey instrument and data collection

The questionnaire was developed as a collaborative process between researchers at Michigan State University (MSU) and the NREMT. A cognitive walk-through (See S1 File, "Cogntive walk-through protocol") was conducted with six participants from the target population to ensure that survey items functioned as intended. Based on this testing, changes were made to enhance understanding by the target population (e.g., the term "medical devices" was changed to "medical supplies").

The questionnaire (see S3 File "Study Questionnaire") included items to characterize respondents (demographic questions and questions regarding their work history) as well as those which gathered information about their experiences with two categories of packaged products: medications and medical supplies. Within each product category, questions were organized to probe difficulties related to two critical tasks: identifying packaged products and opening packages. When respondents reported difficulty identifying or opening a packaged medication or medical supply, a cascading series of questions probed their experience to explore the reasons that they had encountered the difficulty (Objective 1), the way that they coped with the issue (Objective 2), and whether or not it had negatively impacted the patient (Objective 3).

Invitations to take part in the electronic questionnaire were sent to a random sample of 12,000 nationally certified paramedics with valid email addresses. Completion of the questionnaire had no bearing on the person's EMS certification and no identifying information was requested. After the initial invitation, reminder emails were sent approximately one and two weeks later. Respondents who completed a questionnaire by December 1st of 2016, were put into a drawing, with 10 randomly selected to receive $100 Amazon gift cards.

## Data analysis

To be included in the analysis, respondents had to be older than 18, be practicing as a paramedic, have provided medical care in a prehospital setting within the previous 12 months, and have completed the questionnaire in full.

Analyses were conducted using SPSS (IBM, Version 22). Descriptive analysis was performed on frequencies of respondent characteristics as well as their reported reasons for difficulties by critical task (identification and opening) for each of the products (medications and medical supplies). Descriptive variables related to participants were regrouped into meaningful categories to better understand the factors affecting difficulty. Specifically, race was dichotomized to white and all others; years of experience to 10 years or less, and more than 10 years of experience. Education level was categorized as non-college vs college; primary EMS role was categorized as patient care provider and all others (e.g., dispatcher, educator, etc.). Community size was regrouped into the three categories: small town (less than 25,000 residents), medium town (25,000–149,999 people) or city (more than 150,000 people).

We conducted multivariable logistic regression analysis to identify risk factors related to "difficulties associated with opening medication within the past year" and "difficulties

associated with opening medical supplies the past year." Univariable logistic regression was also performed for each independent variable to investigate the marginal relationship between each variable and the binary outcome (had/didn't have difficulty by product category); univariable and multivariable logistic regression analysis were conducted separately for modelling the difficulty in opening medications and the difficulty in opening medical supplies using the variables selected with the previously described method. Odds ratios and 95% confidence intervals (95% CI) were calculated accordingly.

## Results

### Characteristics of study subjects

A total of 1,912 responses were received (response rate = 16%). Respondents were excluded because they had not provided care in the prehospital setting in the past 12 months (n = 193) or due to incomplete responses (n = 17). The majority of the survey respondents were male (79.3%), white (88.7%), had a more than 10 years' experience with EMS (61.5%) and served the role of care provider (74.7%). Table 2 presents an overview of the results related to participant characterization for the 1,702 responses included for analysis.

### Difficulties and coping strategies

Frequencies and proportions related to all three objectives are found in Table 3 (medications) and Table 4 (medical supplies). Nearly 20% of the paramedics indicated that they had

**Table 2. Characteristics of participants.**

| Characteristic | |
|---|---|
| **Age in years, mean (SD)** | **43.2 years (10.6)** |
| **(% of respondents reporting; denominator = 1,702); Frequency (%)** | |
| Sex | |
| Female | 332 (19.5) |
| Male | 1350 (79.3) |
| Missing | 20 (1.2) |
| Race | |
| Non- Hispanic White | 1,509 (88.7) |
| Non-white | 193 (11.3) |
| Years of experience in EMS | |
| 10 years or less than 10 years | 655 (38.5) |
| More than 10 years | 1,047 (61.5) |
| Level of education | |
| Non-College (Did not complete high school and High school graduate/GED) | 87 (5.1) |
| College (Some college, associate degree, Bachelor's degree, Master's degree and Doctoral Degree) | 1,615 (94.9) |
| Primary role | |
| Patient care provider | 1,272 (74.7) |
| Others (educator, preceptor, dispatcher/call taker, administrator/manager, first-line supervisor and others) | 430 (25.3) |
| Size of community | |
| Small town (less than 24,999 people) | 506 (29.7) |
| Medium town (25,000–149,999 people) | 599 (35.2) |
| City size town (more than 150,000 people) | 597 (35.1) |

**Table 3. Frequencies and proportion of respondents self-reporting difficulty with each task (identifying and opening), resulting coping strategies and negative impact on care related to medication use within the previous 12 months.**

| Objective 1- Estimate the prevalence of difficulty associated with the packaging of medications within the prehospital context within the previous 12 months of work | | | |
|---|---|---|---|
| **Difficulty *identifying* a medication 359 respondents indicate difficulty identifying a medication within the past 12 months (21.1%)** | | **Difficulty *opening* a medication 349 respondents indicate difficulty opening a medication within the past 12 months (20.5%)** | |
| **Reasons for difficulty** | n (% of total respondents; % of those reporting this difficulty with this product category) | **Reasons for difficulty** | n (% of total respondents; % of respondents reporting this difficulty with this product category) |
| Lack of transparency made product identification difficult | 47 (2.8; 13.1) | Too small of an area to grip | 125 (7.3; 35.8) |
| Crowded label | 189 (11.1; 52.6) | Material meant to separate stuck together | 116 (6.8; 33.2) |
| Small text | 238 (14.0; 66.3) | Product required too much force to open | 119 (7.0; 34.1) |
| Similar packaging different products | 246 (14.5; 68.5) | Product required two hands to open | 172 (10.1; 49.3) |
| Confusing names | 55 (3.2; 15.3) | Unfamiliar with product packaging | 60 (3.5; 17.2) |
| Dark conditions | 117 (6.9; 32.6) | Packaging directions for opening were not clear | 45 (2.6; 12.9) |
| Objective 2- Investigate the coping strategies employed when difficulties occur with the packaging of medications | | | |
| **Coping strategies** | | **Coping strategies** | |
| Flashlight | 211 (12.4; 58.8) | Knife | 99 (5.8; 28.4) |
| Touch/feel | 26 (1.5; 7.2) | Scissors | 189 (11.1; 54.2) |
| Changed location of product within container, bag or ambulance | 174 (10.2 48.5) | Teeth | 103 (6.1; 29.5) |
| | | Pen | 76 (4.5; 21.8) |
| | | Partner Assist | 172 (10.1; 49.3) |
| Objective 3- Begin to quantify the potential impacts on care associated with difficulties with the packaging of medications | | | |
| **Difficulty resulted in negative patient outcome** | 20 (1.2; 5.6) | **Difficulty resulted in negative patient outcome** | 32 (1.9; 9.2) |

encountered difficulty related to packaging within the previous 12 months for both tasks (identifying and opening) across product categories (medications and medical supplies); specifically: 359 respondents (21.1%) reported that they had difficulty identifying medications within the previous 12 months (Table 3); while 290 respondents (17.0%) reported difficulty identifying medical supplies (Table 4); 349 (20.5%) reported difficulty opening a medication and 399 (23.4%) had difficulty opening a medical supply during this same time frame.

The most commonly reported reason for difficulty identifying a medication was that "different medications have similar packaging" (n = 246, 68.5%, Table 3), while the most commonly reported reason leading to difficulty in identifying a medical supply was "crowded label made it difficult to read" (n = 189, 65.2%, Table 4).

The top reason for difficulty opening was the same across both product categories, "product required two hands to open" (n = 172, 49.3%; n = 270, 67.7%). Frequencies of reported, specific reasons associated with difficulty opening products are presented in Table 3 for medications and Table 4 for medical supplies.

"Flashlight use" was reported as the most common coping strategy employed when paramedics reported difficulty identifying both medications (n = 211/359, 58.8%) and medical supplies (n = 156/290, 53.8%). "Scissors" were the most commonly reported strategy employed when opening difficulties were encountered; this was the case for both medications (n = 189/349, 54.2%; Table 3) and medical supplies (n = 307/399, 76.9%; Table 4). Of additional interest,

**Table 4. Frequencies and proportion of respondents self-reporting difficulty with each task (identifying and opening), resulting coping strategies and negative impact on care related to use of medical supplies within the previous 12 months.**

| Objective 1- Estimate the prevalence of difficulty associated with the packaging of medical supplies with the prehospital context within the previous 12 months of work | | | |
|---|---|---|---|
| **Difficulty identifying a medical supply 290 respondents indicate difficulty identifying a medical supply within the past 12 months (17.0%)** | | **Difficulty opening a medical supply 399 respondents indicate difficulty opening a medical supply within the past 12 months (23.4%)** | |
| **Reasons for difficulty** | n (% of total respondents; % of those reporting this difficulty with this product category) | **Reasons for difficulty** | n (% of total respondents; % of those reporting this difficulty with this product category) |
| Lack of transparency made product identification difficult | 60 (3.5; 20.7) | Too small of an area to grip | 252 (14.8; 63.2) |
| Crowded label | 189 (11.1; 65.2) | Material meant to separate stuck together | 188 (11.0; 47.1) |
| Similar packaging different products | 170 (10.0; 58.6) | Product required too much force to open | 113 (6.6; 28.3) |
| Confusing names | 20 (1.2; 6.9) | Product required two hands to open | 270 (15.9; 67.7) |
| Dark conditions | 88 (5.2; 30.3) | Unfamiliar with product packaging | 33 (1.9; 8.3) |
| | | Packaging directions for opening were not clear | 51 (3.0; 12.8) |
| Objective 2- Investigate the coping strategies employed when difficulties occur with the packaging of medical supplies | | | |
| **Coping strategies** | | **Coping strategies** | |
| Flashlight | 156 (9.2; 53.8) | Knife | 143 (8.4; 35.8) |
| Touch/feel | 73 (4.3; 25.2) | Scissors | 307 (18.0; 76.9) |
| Changed location of product within container, bag or ambulance | 144 (8.5; 49.7) | Teeth | 156 (9.2; 39.1) |
| | | Pen | 96 (5.6; 24.1) |
| | | Partner Assist | 219 (12.9; 54.9) |
| Objective 3- Begin to quantify the potential impacts on care when difficulties occur with the packaging of medical supplies | | | |
| **Difficulty resulted in negative patient outcome** | 31 (1.8; 10.7) | **Difficulty resulted in negative patient outcome** | 51 (3.0; 12.8) |

was the reported use of teeth to open both medications (n = 103/349, 29.5%) and medical supplies (n = 156/399, 39.1%) when difficulties were encountered.

We fit two multivariable logistic regression models with the two outcomes of interest: difficulty opening medication (Model 1, Table 5) and difficulty opening medical supplies (Model 2, Table 6). The following covariates were considered for both models: age, sex, race, years of experience in EMS, primary role, the level of education, and size of community. In addition, difficulty in identifying medication and difficulty in identifying medical supplies were included for Model 1 and Model 2, respectively.

Tables 4 and 5 present the logistic regression analysis for difficulties opening medications and difficulties opening medical supplies, respectively. The odds of difficulty opening medications for those who had experienced difficulty identifying medications within the past year was 2.91 times that of the odds of difficulty opening medications for those who had not experienced difficulty identifying medications within the past year (OR: 2.91, 95% CI: 2.21–3.83). Significant effects of sex and primary role on the odds of reporting difficulty opening medications were also observed (Table 5). Specifically, the odds of reporting a difficulty opening medication was reduced by approximately 49% for males when compared to that of female respondents (OR: 0.51, 95% CI: 0.38–0.68). Additionally, the odds of reporting difficulty with opening medications were reduced by 29% for those with other roles such as dispatcher,

**Table 5. Univariable and multivariable regression analysis for difficulty opening medication.**

| | Univariable analysis | | | Multivariable analysis | | |
|---|---|---|---|---|---|---|
| | Odds Ratio | 95% C.I | | Odds Ratio | 95% C.I | |
| **Difficulty in identifying medication** | | | | | | |
| Did not have difficulty in identifying medication within the past 12 months | **ref** | | | **ref** | | |
| Had difficulty in identifying medication within the past 12 months | **2.70** | **2.081** | **3.51** | **2.91** | **2.21** | **3.83** |
| **Age** | 0.99 | 0.98 | 1.00 | 0.99 | 0.98 | 1.0070 |
| **Sex** | | | | | | |
| Female | **ref** | | | **ref** | | |
| Male | **0.53** | **0.41** | **0.70** | **0.51** | **0.38** | **0.68** |
| **Race/ethnicity** | | | | | | |
| Non-Hispanic white | ref | | | ref | | |
| Minority | 0.81 | 0.54 | 1.20 | 0.69 | 0.43 | 1.10 |
| **Years of EMS experience** | | | | | | |
| = < 10 years | ref | | | ref | | |
| >10 years | 0.85 | 0.66 | 1.076 | 0.98 | 0.70 | 1.37 |
| **Primary role** | | | | | | |
| Patient care provider | **ref** | | | **ref** | | |
| Others (educator, preceptor, dispatcher/call taker, administrator/manager, first-line supervisor and others) | **0.68** | **0.50** | **0.91** | **0.71** | **0.52** | **0.99** |
| **Level of education** | | | | | | |
| Non-college (didn't complete high school and high school graduate/GED) | ref | | | ref | | |
| College (some college, associate degree, bachelor's degree, master's degree and doctoral degree) | 0.89 | 0.53 | 1.50 | 0.86 | 0.49 | 1.49 |
| **Size of community** | | | | | | |
| Small town (less than 24,999 people) | ref | | | ref | | |
| Medium town (25,000–149,999 people) | 1.066 | 0.80 | 1.43 | 1.12 | 0.82 | 1.54 |
| City size town (more than 150,000 people) | 0.92 | 0.69 | 1.23 | 0.94 | .69 | 1.27 |

Bolded font indicates significance at α = 0.05.

instructor and administrator, compared to patient care providers (OR: 0.71, 95% CI: 0.52–0.97).

When difficulties associated with opening medical supplies were examined (Table 6), results were similar to those reported for opening medications (Table 5). The odds of difficulty opening medical supplies for participants who reported a difficulty identifying medical supplies within the past 12 months were 3.67 times higher than the odds for those who had not reported difficulty identifying a medical supply within the past 12 months (OR: 3.67, 95% CI: 2.80–4.85). For every one-year increase in age, there was a 2% increase in the odds of difficulty opening medical supplies (OR: 1.017, 95% CI: 1.00–1.03).

## Negative patient outcomes

The prevalence of a negative outcome to patient care outcomes affiliated with difficulties with packaging ranged from 1.2% of all participants (n = 20, or 5.6% of the 359 people reporting difficulty identifying a medication, Table 3) to as high as 3.0% of all participants (n = 51, or 12.8% of the 399 people indicating that they had encountered difficulty opening a medical supply, Table 4). Intermediate to these two task/product combinations were those related to identifying a medical supply (Table 4), with 31 paramedics reporting a negative outcome on patient care (1.8% of respondents or 10.7% of the 290 who reported experiencing this difficulty) and 32 negative outcomes associated with opening medications (1.9% of the respondent population and 9.2% of 349 reporting difficulty with this task, Table 3).

**Table 6. Univariable and multivariable regression analysis for difficulty opening medical supplies.**

| | Univariable analysis | | | Multivariable analysis | | |
|---|---|---|---|---|---|---|
| | Odds Ratio | 95% C.I | | Odds Ratio | 95% C.I | |
| **Difficulty in identifying medical supplies** | | | | | | |
| Did not have difficulty in identifying medical supplies within the past 12 months | **ref** | | | **ref** | | |
| Had difficulty in identifying medical supplies within the past 12 months | **3.45** | **2.64** | **4.51** | **3.67** | **2.78** | **4.85** |
| **Age** | 1.0050 | 0.99 | 1.016 | **1.017** | **1.0020** | **1.031** |
| **Sex** | | | | | | |
| Female | ref | | | ref | | |
| Male | 0.99 | 0.74 | 1.31 | 0.90 | 0.67 | 1.21 |
| **Race/ethnicity** | | | | | | |
| Non-Hispanic white | ref | | | ref | | |
| Minority | 0.81 | 0.56 | 1.17 | 0.71 | 0.47 | 1.078 |
| **Years of EMS experience** | | | | | | |
| = < 10 years | ref | | | ref | | |
| >10 years | 1.025 | 0.81 | 1.29 | 0.83 | 0.61 | 1.14 |
| **Primary role** | | | | | | |
| Patient care provider | ref | | | ref | | |
| Others (educator, preceptor, dispatcher/call taker, administrator/manager, first-line supervisor and others) | 0.99 | 0.76 | 1.28 | 1.023 | 0.77 | 1.36 |
| **Level of education** | | | | | | |
| Non-college (didn't complete high school and high school graduate/GED) | ref | | | ref | | |
| College (some college, associate degree, bachelor's degree, master's degree and doctoral degree) | 1.38 | 0.79 | 2.40 | 1.53 | 0.84 | 2.79 |
| **Size of community** | | | | | | |
| Small town (less than 24,999 people) | ref | | | ref | | |
| Medium town (25,000–149,999 people) | 0.83 | 0.62 | 1.090 | 0.82 | 0.61 | 1.11 |
| City size town (more than 150,000 people) | 0.84 | 0.64 | 1.091 | 0.86 | 0.65 | 1.14 |

Bolded font indicates significance at α = 0.05.

## Discussion

Although other researchers have indicated healthcare products not designed in ways which consider care context as among the most common hazard associated with medical errors [23], there is a dearth of work characterizing how packaged products perform in realistic contexts of use. The most oft-studied healthcare environment is the perioperative context, an environment that is arguably less demanding for product design than that of prehospital environment. Further, the performance of package design, a critical and ubiquitous component of all products, has, to our knowledge, never been investigated in prehospital environments, despite the fact that similar packaging and labelling have been identified as fundamental reasons for medication errors [27].

Our analysis provides some initial evidence that paramedics have difficulties both identifying and opening packaged products and that these difficulties have the potential to negatively impact patient outcomes (Tables 3 and 4). Several factors were found to be significantly associated with the reported difficulty with opening for both medications and medical supplies. For both categories of product (medications and medical supplies), respondents had significantly higher odds of reporting difficulty with opening if they had also reported difficulty with identifying within the past year (Tables 5 and 6). These results could indicate that respondents who were more comfortable reporting difficulties were more likely to report for each of the tasks studied (identifying and opening); however, it could also suggest that when manufacturers do

not engage in a thoughtful design process, poor design affects multiple, critical tasks. If the latter is the case, paramedics could encounter serial difficulties among the tasks to be accomplished (first in identifying and then, subsequently, in opening needed products), potentially delaying patient care. The potential and impact for serialized difficulties associated with packaging in prehospital contexts was beyond the scope of the work but presents an interesting idea that needs study.

Results also advocate for the incorporation of users likely to have difficulties associated with packaging during the iterative design process. Findings indicate that females were significantly more likely to report difficulty opening medications than their male counterparts (Table 5), potentially because of documented differences in hand strength [28–31]. That said, this could also be indicative of a willingness to express vulnerability that men do not share [32]; if this is the case, it would support the idea that difficulties are actually underreported. We also found older paramedics to be significantly more likely to report difficulties opening medical supplies than their younger counterparts (Table 6); declines in the strength of the hand are well-documented as a natural process that occurs with aging [29,33] and offer an explanation of this result.

The limited work that has been done to understand how packaging performs in healthcare contexts has focused on *perioperative environments* [27,34,35]. Many of the published findings conducted in these more controlled contexts of care indicate improvements in packaging are needed, parallel to the findings we present here. Published studies using perioperative contexts have concluded that packages are: "hard to open" [35,36] and employ "crowded label contents" which make crucial information difficult to locate [27,34,35]. While our survey of paramedics echoed the difficulties voiced by perioperative personnel, they also expressed problems, and coping strategies, that were uniquely their own. The indication or implication that paramedics only have one hand available to physically manipulate packaging was apparent throughout the results. The need for designs which enable one-handed use was evident in this sample, with the "need for two hands [when opening the packages] [32]" reported as a common difficulty and "use of teeth" or the need for a "partner's assistance" as common coping mechanisms when difficulties were encountered. It is not beyond fathom that the use of instruments or teeth have the potential to serve as an indirect mechanism for the transfer of microbes, one possible contributor among many which could help to explain the high rates of infection among patients who have received advanced life support (ALS) prior to hospital admission as compared to those transported by other means [25].

If this is the case, it would suggest that the rates of negative impact on patient outcome (Tables 3 and 4) are actually under-reported, further supporting this as an issue in need of careful consideration.

## Limitations

One of the more interesting contributions of our study is the number of participants who reported having difficulties that resulted in negative impact on patient care (1.2% associated with identifying a medication; 1.9% associated with opening a medication;1.8% associated with identifying a medical supply; and 3% associated with opening a medical supply). Readers are urged to interpret results related to negative patient outcomes cautiously. The survey itself did not specifically define the term "negative outcome," instead leaving the survey respondent to interpret its meaning (see Supplemental files for survey). The open-ended nature of this approach enables a wide array of potential outcomes to be considered as negatively impacting care (from delay of care to death of patient). A lack of questions that probed details

surrounding the negative outcome precludes detailed interpretation of these results and implores the need for future research.

A further limitation of the survey is the binary nature of the question about difficulties affiliated with packaging use. Specifically, a single question asks respondents about difficulties (identifying or opening) by product type (medications or medical device) and is only presented once (for each combination). A "yes response" related to the difficulty question triggered a cascade of questions probing the difficulty and coping strategies that were employed. Although this provides us with an indication of the frequency across participants, the way that the survey was framed we cannot deduce the number of occurrences for each respondent (across the last year) or whether or not the coping mechanisms that they reported arose from multiple incidences or a single event.

## Conclusions

Our findings suggest that paramedics encountered difficulties identifying and opening packaged products and provides preliminary evidence that suboptimal designs potentially negatively impact patient outcomes. In light of this, the designers of healthcare products should incorporate insights gathered in prehospital environments to enhance package designs in ways that enable the performance of tasks critical to care. This becomes an imperative as healthcare evolves to better integrate the complex components that must work in synergy toward common, patient-centered goals, and is requisite to improved outcomes [37].

## Supporting information

**S1 File. "Cognitive walk-through protocol".**
(DOCX)

**S2 File. "Flat file data set".**
(XLSX)

**S3 File. "Study questionnaire".**
(PDF)

## Acknowledgments

The author would like to thank Hope Akaeze who assisted in her role with the MSU CSTAT organization. Ms. Akaeze provided useful resources for statistics concepts presented herein.

## Author Contributions

**Conceptualization:** Jiyon Lee, Laura Bix.

**Data curation:** Jiyon Lee, Rebecca E. Cash, Remle P. Crowe, Ashish R. Panchal, Laura Bix.

**Formal analysis:** Jiyon Lee, Hyokyoung G. Hong, Laura Bix.

**Funding acquisition:** Laura Bix.

**Investigation:** Jiyon Lee, Rebecca E. Cash, Remle P. Crowe, Ashish R. Panchal, Laura Bix.

**Methodology:** Jiyon Lee, Laura Bix.

**Project administration:** Jiyon Lee, Laura Bix.

**Resources:** Rebecca E. Cash, Remle P. Crowe, Ashish R. Panchal, Kami Silk, Marvin Helmker, Laura Bix.

**Supervision:** Laura Bix.

**Validation:** Jiyon Lee, Rebecca E. Cash, Remle P. Crowe, Ashish R. Panchal, Laura Bix.

**Visualization:** Jiyon Lee, Laura Bix.

**Writing – original draft:** Jiyon Lee, Laura Bix.

**Writing – review & editing:** Jiyon Lee, Rebecca E. Cash, Remle P. Crowe, Hyokyoung G. Hong, Ashish R. Panchal, Kami Silk, Marvin Helmker, Laura Bix.

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
