## [Decision Letter · Decision Letter 0]

19 May 2021

PONE-D-20-33281

Paramedic interactions with the packaging of medications and medical supplies in the prehospital setting: Does poor package design have the potential to impact patient health?

PLOS ONE

Dear Dr. Bix,

Thank you for submitting your manuscript to PLOS ONE. After careful consideration, we feel that it has merit but does not fully meet PLOS ONE’s publication criteria as it currently stands. Therefore, we invite you to submit a revised version of the manuscript that addresses the points raised during the review process.

When preparing your revision, please pay particular attention to clarifying the aspects of your study indicated by the reviewers in their comments.

We look forward to receiving your revised manuscript.

Kind regards,

Jamie Males

Staff Editor

PLOS ONE

Journal Requirements:

3. Please include additional information regarding the content validation of the survey or questionnaire used in the study and ensure that you have provided sufficient details that others could replicate the analyses. Furthermore if the questionnaire is not under a copyright more restrictive than CC-BY, please include a copy, in both the original language and English, as Supporting Information.

Reviewers' comments:

Reviewer's Responses to Questions

**Comments to the Author**

1. Is the manuscript technically sound, and do the data support the conclusions?

Reviewer #1: Yes

Reviewer #2: Partly

Reviewer #3: Yes

2. Has the statistical analysis been performed appropriately and rigorously? 

Reviewer #1: Yes

Reviewer #2: Yes

Reviewer #3: Yes

3. Have the authors made all data underlying the findings in their manuscript fully available?

Reviewer #1: Yes

Reviewer #2: Yes

Reviewer #3: Yes

4. Is the manuscript presented in an intelligible fashion and written in standard English?

Reviewer #1: Yes

Reviewer #2: Yes

Reviewer #3: Yes

5. Review Comments to the Author

Reviewer #1: Thanks the opportunity to review your work. The study is relevant to daily clinical practice of prehospital patient care. The study is well designed and conducted. The result and date analysis are appropriate to address the research questions.

Some questions regarding "patient outcome" in Table 2. That is very vague definition in healthcare setting. Does it mean the patient care delayed, or infection, or patient death? How could those responders know which packing factors contribute those outcomes? This could introduce some bias.

Most of those paramedics working as a team, it might be interesting to know how their partner helped them to complete those tasks with difficulty (e.g., one hand). Also, beyond the lighting, does the moving vehicle compare to stable task environment contribute more challenge for their tasks.

Those are issue can be mentioned in the discussion to help interpret those results and provide guidance for future packing design.

Reviewer #2: The authors study the usability of packaging and medical supplies in the pre-hospital setting.

This is a rather straightforward paper. There are a few points that require additional explanation, and there is one major reflection regarding the study set-up:

- 'pre-hospital' should be more clearly defined.

- p3: what does it actually mean "of increasing importance"?

- p10: upper age limit of 85 seems a bit weird? If that high, why have one at all?

- p11: "descriptive variables (...) the factors affecting difficulty". How can reducing the granularity of your data lead to better understanding? Did you do a sensitivity analysis for the years of experience?

Major point:

- the survey questions are not appended, causing some unclarity. Hence, this point is articulated into multiple tracks. Please clarify better in your text which is the actual scenario, and address the points presented for that scenario.

The authors present extensive quantitative analysis of the results. However, it is unclear what these actually mean. I would image that in the group of respondents there are A.) people not experiencing problems (or not remembering or not willing to answer), B.) respondents remembering a single instance of an issue in the past year, C.) respondents who had experienced multiple issues.

My unclarity concerns group C.

Option 1: the survey asked respondents to repeat the questions for multiple instances, so one respondent could generate multiple recorded cases and coping mechanisms separately.

Option 2: respondents answered only once, but could include all coping mechanisms (over multiple issues in the last year)

Option 3: respondents could answer only regarding a single specific case, and were or were not prompted on how to select the case for which they provided the answers.

Which of these options is the actual case, and what that implies for the presented statistics is currently unclear.

The current discussion is already a well written reflection of what the results might mean, but they should be extended to reflect on the above points raised regarding the sampling of problematic interactions.

Reviewer #3: The title of this paper need to change. Please restructure the title. Better remove ? on the title.

Please make a proof ready on this paper.

Please do the proof read on this paper to make more technical and easy to understanding for the reader.

6. PLOS authors have the option to publish the peer review history of their article (what does this mean?). If published, this will include your full peer review and any attached files.

Reviewer #1: No

Reviewer #2: No

Reviewer #3: No

---

## [Author Response · Author response to Decision Letter 0]

28 Jun 2021

[EDITOR] 1. Please ensure that your manuscript meets PLOS ONE's style requirements, including those for file naming. The PLOS ONE style templates can be found at

[AU]—We have thoroughly reviewed the style templates and believe our submission to be compliant with the requirements of the publication. 

[EDITOR] 2. Please review your reference list to ensure that it is complete and correct. If you have cited papers that have been retracted, please include the rationale for doing so in the manuscript text, or remove these references and replace them with relevant current references. Any changes to the reference list should be mentioned in the rebuttal letter that accompanies your revised manuscript. If you need to cite a retracted article, indicate the article’s retracted status in the References list and also include a citation and full reference for the retraction notice.

[EDITOR] 3. Please include additional information regarding the content validation of the survey or questionnaire used in the study and ensure that you have provided sufficient details that others could replicate the analyses. Furthermore if the questionnaire is not under a copyright more restrictive than CC-BY, please include a copy, in both the original language and English, as Supporting Information.

[AU]--- We have now included both the survey, which is not currently under copyright, and the protocol for the cognitive walkthrough that was conducted with a small sample of practicing Emergency Medical Service (EMS) professionals prior to distribution of the survey. These are included as supplemental materials. The cognitive walk through was conducted to examine the survey content for applicability with test population by conatively testing practicing EMS providers. Specifically, we examined whether survey items functioned as intended. There were four main areas to ascertain; 

(1) Is the item being comprehended as intended by the item writer?

(2) Is the respondent a knowledgeable informant? That is, does the respondent possess the knowledge needed to answer the question?

(3) Can respondents use the knowledge they possess to form a judgment (answer the question) in an appropriate manner?

(4) Will respondents with similar experiences select the same response option?

Six EMS professionals participated in the test. The participants included: an EMT in an ALS level service with 6 years of experience, an EMT at a fire-based service with unknown years of experience, a paramedic at a fire-based service with 12 years of experience, a paramedic at a fire-based 911 service with 8 years of experience, a paramedic at a private service with 5 years of experience, and a critical care paramedic at a private service with 7 years of experience. 

As a result of this assessment, changes to the survey were recommended, and reflected in a revised survey instrument that was distributed nationally. 

The validity of sample size is addressed in the paper, which was calculated by performing power calculation. 

[AU] The survey (as well as the cognitive walkthrough) is now included, in its entirety, as part of the supplemental materials. Captions for each of these have been added to the revised document. 

Reviewers' comments:

Reviewer's Responses to Questions

Comments to the Author

1. Is the manuscript technically sound, and do the data support the conclusions?

Reviewer #1: Yes

Reviewer #2: Partly

Reviewer #3: Yes

2. Has the statistical analysis been performed appropriately and rigorously? 

Reviewer #1: Yes

Reviewer #2: Yes

Reviewer #3: Yes

3. Have the authors made all data underlying the findings in their manuscript fully available?

Reviewer #1: Yes

Reviewer #2: Yes

Reviewer #3: Yes

4. Is the manuscript presented in an intelligible fashion and written in standard English?

Reviewer #1: Yes

Reviewer #2: Yes

Reviewer #3: Yes

5. Review Comments to the Author

Reviewer Comments presented in bold--- Author response preceded by [AU]

[REVIEWER #1]: Thanks the opportunity to review your work. The study is relevant to daily clinical practice of prehospital patient care. The study is well designed and conducted. The result and date analysis are appropriate to address the research questions.

[AU] We thank the reviewer for their kind words. 

[REVIEWER #1]-Some questions regarding "patient outcome" in Table 2. That is very vague definition in healthcare setting. Does it mean the patient care delayed, or infection, or patient death? How could those responders know which packing factors contribute those outcomes? This could introduce some bias.

[AU] --- The authors appreciate the insight provided by the reviewer regarding lack of specificity of the term “patient outcome.” The reviewer is astute in this observation. As we indicate in the introduction of the paper, errors associated with packaging and labeling tend to be latent in nature; that is, they occur upstream from the presentation. Adding to the complexity is the fact that they tend to be “system related,” specifically, that they are one of a number of contributing factors. These facts suggest that any numbers reported in the publication herein are likely to be conservative estimates of the magnitude of the problem. That said, after reviewing the comments from the reviewer, we realize and appreciate that a negative patient outcome is not necessarily related to health status, but, as noted, could include other things, such as delayed care. The inclusion of the survey as a supplemental file will allow the reviewer to see that little can be done regarding how the outcome language was framed, leaving a very broad potential for interpreting the term. 

The lack of the specificity of the outcomes reported, are, indeed, a shortcoming of the study design. To more clearly and directly address this with readers, we have added a limitations section to the document, a section which was not included in the prior version. We have incorporated the following text as part of the limitations.

“One of the more interesting contributions of our study is the number of participants who reported having difficulties that resulted in negative impact on patient care (1.2% associated with identifying a medication; 1.8% associated with identifying a medical supply; 3% associated with opening a medical supply). Readers are urged to interpret results related to negative patient outcomes cautiously. The survey itself did not specifically define what encompassed a “negative outcome,” instead leaving the survey respondent to interpret its meaning (see Supplemental files for survey). The open-ended nature of this approach enables a wide array of potential outcomes to be considered as negatively impacting care (from delay of care to death). The lack of questions that probed details surrounding the negative outcome precludes detailed interpretation of these results and implores the need for future research.” 

[REVIEWER #1]-Most of those paramedics working as a team, it might be interesting to know how their partner helped them to complete those tasks with difficulty (e.g., one hand). Also, beyond the lighting, does the moving vehicle compare to stable task environment contribute more challenge for their tasks….[REVIEWER #1]-Those are issue can be mentioned in the discussion to help interpret those results and provide guidance for future packing design.

[AU]- We believe that the inclusion of the survey (now included in supplemental materials) will significantly aid reader understanding. 

The survey presented “partner assistance” as one of the radio buttons that could be checked (multiple radio buttons could be checked) as a coping strategy utilized. Although an open-ended response was provided (under other), the wording did not likely lead respondents to elaborate on the type of assistance that they requested/obtained. In retrospect, a more granular collection related to frequencies of specific events (and details associated with the same) would have provided interesting insights. 

That said, the overarching goal of this work was to unearth whether or not packaging used within the prehospital context was an area in need of further study. Additionally, we wanted to verify /validate the anecdotal evidence that we had which suggested problems associated with packaging in prehospital contexts. Our work does provide evidence of this need beyond the anecdotes and, as such, we believe there is still value in this publication. 

With regard to other factors that influence care delivery (such as vibration), that is, precisely, what we were interested in studying. We had observed that paramedics used their teeth to open packaging, largely due to the extremes contexts that they have to deal with but could find no indication of this within the literature. Our review of the literature suggested no characterization of packaging shortcomings in the prehospital context. Further, our experience with packaged healthcare products suggested that, in the rare case that human factors studies were done with packaging, the sole focus was on perioperative environments. After conducting this survey to confirm and validate our observations, we ran simulation studies with paramedics which incorporated motion (using a multi access vibration table driven with profiles taken from road data collected using two ambulances); this data replicates many of the findings presented here, including use of teeth to open. We are currently writing up the results of the study, which did replicate the problematic behaviors noted here. In short, vibration is one of many factors that catalyzes some of the behaviors. 

[REVIEWER #1]: The authors study the usability of packaging and medical supplies in the pre-hospital setting.

This is a rather straightforward paper. There are a few points that require additional explanation, and there is one major reflection regarding the study set-up:

[REVIEWER #1]- 'pre-hospital' should be more clearly defined.

[AU] We have made an attempt to clarify this terminology, jargon of the healthcare sector, that we failed to define in our first submission. 

In the abstract we have amended the very first section (Background) so that it now reads 

“Settings where Emergency Medical Services (EMS) are provided to stabilize patients and transport them to locations better equipped to provide comprehensive care, “prehospital settings,” are not frequently considered when designing packaged products. More specifically, packaging design is an understudied area in the prehospital setting, potentially impacting both healthcare provider behavior and patient outcomes. Our objectives were to: 1) describe difficulties associated with packaging in prehospital settings 2) investigate the coping strategies used by paramedics when difficulties occurred, and 3) assess the potential impacts these difficulties had on patient care. 

Within the body of the document, just prior to the Methods section- at the tail end of the literature review, we have added the following paragraph:

“And it isn’t just packaging that hasn’t been thoroughly researched, understood, and optimized; the healthcare environment itself, undoubtedly, impacts how care givers interact with products to deliver patient care. Of particular interest to us was how packaging performs within the prehospital setting. Emergency Medical Services (EMS) are administered during prehospital care to stabilize patients and transport them to a location better equipped to provide comprehensive care. Despite the extreme conditions that may be present (e.g. poor lighting, extreme heat, noise, chaos, emergency vehicle movement, interloping friends and family), they are infrequently (if ever) considered by designers.”

[REVIEWER #1] -p3: what does it actually mean "of increasing importance"?

[AU]- To address this with better specificity, the sentence “Engineering safety into the overall healthcare system to reduce errors and improve patient outcomes is of increasing importance.” (appearing at the very beginning of the article) has now been modified to the following text in an attempt to enhance the clarity of message. 

“Engineering safety into the overall health care system to reduce errors and improve patient outcomes is an important paradigm being actively embraced by the designers of healthcare products (and systems), caregivers and policy makers [1].

[REVIEWER #1]- p10: upper age limit of 85 seems a bit weird? If that high, why have one at all?

[AU] We concur that this is a bit odd, and in retrospect, that we probably just should have been indicated “ no upper limit” regarding age. That said, the paperwork filed for our IRB requires an age range for eligible participants (with upper and lower limits). The minimum age that someone can consent themselves, and the minimum age to work as a paramedic, is 18 years old. The physical nature of the work skews demographic data young. Specifically, the average age of a female paramedic is 34.7, and the average age of male paramedics is 36.5. Although there is no official limit on the age of people that can work as paramedics, data suggests that there are no paramedics that work beyond their sixties (see below-- Source data from the US Census Bureau ACS PUMS available at https://datausa.io/profile/soc/emergency-medical-technicians-paramedics )

As a result, the upper limit of 85 basically afforded us inclusion of all working paramedics. 

[REVIEWER #1]- p11: "descriptive variables (...) the factors affecting difficulty". How can reducing the granularity of your data lead to better understanding? Did you do a sensitivity analysis for the years of experience?

[AU]- We appreciate and concur with the reviewer’s assessment that collapsing potential independent variables across categories (See Survey in Supplemental files for details that were collected from survey respondents) doesn’t enhance granularity, but a small number of responses for a given category within the independent variable precluded our ability to draw meaningful inference. As such, several categories that we collected were collapsed. 

To further investigate this (in response to reviewer concerns) a sensitivity analysis was conducted with different cutoff points (4 years, 7 years and 15 years) for each of the two products (medications and medical supplies). These analyses are presented in the tables below and suggest limited changes in significant results regardless of the cutoff points employed. 

 4 years cutoff 7 years cutoff 15 years cutoff

 =< cutoff point 210 (12.3%) 438 (25.7%) 980 (57.6%)

> cutoff point 1,491 (87.7%) 1,263 (74.3%) 721 (42.4%)

Missing 1 1 1

Total 1,702 1,702 1,702

The results of logistic regression per cutoff (4 years, 7 years and 15 years) within the variable of ‘years of experience’ are shown in the tables presented below. These were conducted by product (1. Medication and 2. Medical supplies).

1. For ‘difficulty opening medication’, the current (included in the published paper) cutoff of group (10 years) indicates that the coefficients of ‘difficulty identifying medication’, ‘sex’ and ‘primary role’ as significant. This is consistent with the results of other cutoff analyses that we conducted in response to reviewer concerns (see Table 1 -3).

1-1 Difficulty opening medication: Cutoff at 4 years of experience ….

Variables in the Equation

 B S.E. Wald df Sig. Exp(B)

Step 1a In the past 12 months, have you had difficulty identifying a medication while providing care in the prehospital setting? 1.063 .141 56.991 1 .000 2.896

 Age -.005 .007 .506 1 .477 .995

 What is your sex? -.672 .149 20.260 1 .000 .511

 Race -.375 .240 2.441 1 .118 .687

 year4 -.359 .197 3.315 1 .069 .699

 Primary Role -.313 .164 3.643 1 .056 .732

 Education -.157 .283 .306 1 .580 .855

 Community SIze 1.453 2 .484 

 Community SIze(1) -.189 .161 1.383 1 .240 .828

 Community SIze(2) -.135 .161 .708 1 .400 .874

 Constant .059 .491 .015 1 .904 1.061

1-2. Difficulty opening medication: Cutoff at 7 years of experience ….

Variables in the Equation

 B S.E. Wald df Sig. Exp(B)

Step 1a In the past 12 months, have you had difficulty identifying a medication while providing care in the prehospital setting? 1.070 .141 57.754 1 .000 2.916

 Age -.006 .007 .783 1 .376 .994

 What is your sex? -.678 .149 20.715 1 .000 .508

 Race -.374 .240 2.437 1 .119 .688

 year7 -.135 .172 .618 1 .432 .874

 Primary Role -.321 .165 3.786 1 .052 .725

 Education -.157 .283 .309 1 .578 .854

 Community SIze 1.259 2 .533 

 Community SIze(1) -.178 .160 1.226 1 .268 .837

 Community SIze(2) -.117 .160 .540 1 .463 .889

 Constant -.308 .440 .491 1 .483 .735

1-3. Difficulty opening medication: Cutoff at 15 years of experience ….

Variables in the Equation

 B S.E. Wald df Sig. Exp(B)

Step 1a In the past 12 months, have you had difficulty identifying a medication while providing care in the prehospital setting? 1.066 .141 57.383 1 .000 2.904

 Age -.009 .008 1.304 1 .254 .991

 What is your sex? -.680 .149 20.773 1 .000 .506

 Race -.375 .240 2.448 1 .118 .687

 year15 -.003 .177 .000 1 .985 .997

 Primary Role -.341 .165 4.246 1 .039 .711

 Education -.154 .283 .295 1 .587 .858

 Community SIze 1.298 2 .522 

 Community SIze(1) -.181 .160 1.277 1 .259 .834

 Community SIze(2) -.115 .160 .514 1 .473 .892

 Constant -.413 .421 .964 1 .326 .662

2. We repeated this exercise related to medical device results. Specifically the analysis that we present in the paper regarding the difficulty opening medical devices suggest that the coefficients of ‘difficulty identifying medical device’ and ‘age’ as significant. In the analysis conducted in response to reviewer concerns these are consistent across the other cutoffs except for the result from 15 years of cutoff (where the coefficient of ‘Age’ is not significant). We postulate that this is because the age is correlated to ‘years of experience’ 

2-1. Difficulty opening medical device: Cutoff at 4 years of experience ….

Variables in the Equation

 B S.E. Wald df Sig. Exp(B)

Step 1a In the past 12 months, have you had difficulty identifying a medical supply (e g , syringe, endotracheal tube, IV administration set) while providing care in the prehospital setting? 1.293 .142 83.202 1 .000 3.644

 Age .016 .006 6.191 1 .013 1.016

 What is your sex? -.100 .152 .437 1 .508 .905

 Race -.344 .214 2.592 1 .107 .709

 year4 -.349 .193 3.254 1 .071 .706

 Primary Role .017 .144 .014 1 .906 1.017

 Education .429 .307 1.951 1 .162 1.535

 Community SIze 1.732 2 .421 

 Community SIze(1) .031 .153 .040 1 .842 1.031

 Community SIze(2) .181 .152 1.424 1 .233 1.199

 Constant -1.874 .503 13.912 1 .000 .153

2-2. Difficulty opening medical device: Cutoff at 7 years of experience ….

Variables in the Equation

 B S.E. Wald df Sig. Exp(B)

Step 1a In the past 12 months, have you had difficulty identifying a medical supply (e g , syringe, endotracheal tube, IV administration set) while providing care in the prehospital setting? 1.291 .142 83.023 1 .000 3.635

 Age .015 .007 4.850 1 .028 1.015

 What is your sex? -.106 .151 .492 1 .483 .899

 Race -.341 .213 2.554 1 .110 .711

 year7 -.159 .165 .929 1 .335 .853

 Primary Role .015 .145 .011 1 .917 1.015

 Education .425 .307 1.917 1 .166 1.529

 Community SIze 1.991 2 .369 

 Community SIze(1) .042 .153 .075 1 .784 1.043

 Community SIze(2) .197 .151 1.703 1 .192 1.218

 Constant -2.215 .454 23.840 1 .000 .109

2-3. Difficulty opening medical device: Cutoff at 15 years of experience ….

Variables in the Equation

 B S.E. Wald df Sig. Exp(B)

Step 1a In the past 12 months, have you had difficulty identifying a medical supply (e g , syringe, endotracheal tube, IV administration set) while providing care in the prehospital setting? 1.287 .142 82.648 1 .000 3.620

 Age .013 .007 3.086 1 .079 1.013

 What is your sex? -.105 .152 .483 1 .487 .900

 Race -.340 .213 2.545 1 .111 .711

 year15 -.048 .162 .090 1 .764 .953

 Primary Role -.001 .145 .000 1 .992 .999

 Education .430 .307 1.965 1 .161 1.537

 Community SIze 2.060 2 .357 

 Community SIze(1) .038 .153 .060 1 .806 1.038

 Community SIze(2) .199 .151 1.724 1 .189 1.220

 Constant -2.342 .434 29.076 1 .000 .096

[REVIEWER #2] Major point:

[REVIEWER #2]- the survey questions are not appended, causing some unclarity. Hence, this point is articulated into multiple tracks. Please clarify better in your text which is the actual scenario, and address the points presented for that scenario.

[AU] The survey is now included in the supplemental materials. We anticipate that this will alleviate several of the issues raised by reviewers. In addition, we have attempted to modify the manuscript to clarify as well. We have addressed these in the specific points throughout this letter. 

[REVIEWER #2] The authors present extensive quantitative analysis of the results. However, it is unclear what these actually mean. I would image that in the group of respondents there are A.) people not experiencing problems (or not remembering or not willing to answer), B.) respondents remembering a single instance of an issue in the past year, C.) respondents who had experienced multiple issues.

My unclarity concerns group C.

Option 1: the survey asked respondents to repeat the questions for multiple instances, so one respondent could generate multiple recorded cases and coping mechanisms separately.

Option 2: respondents answered only once, but could include all coping mechanisms (over multiple issues in the last year)

Option 3: respondents could answer only regarding a single specific case, and were or were not prompted on how to select the case for which they provided the answers.

Which of these options is the actual case, and what that implies for the presented statistics is currently unclear.

The current discussion is already a well written reflection of what the results might mean, but they should be extended to reflect on the above points raised regarding the sampling of problematic interactions.

[AU]- It was clear from the comments of several of the reviewers (and, honestly, should have been self-apparent) that the absence of the original survey made it difficult to completely understand the Methods (and, as such, interpret the Results). We believe that the inclusion of the survey as a supplemental document significantly enhances reader understanding. That said, several of the responses within the survey were “cascading.” That is, subsequent questions would only be shown if a particular response (e.g., a “yes” associated with the question of difficulty within the previous twelve months triggered probing questions related to specific details regarding the difficulty and how participants coped with it). 

The reviewer makes an astute observation; we treated difficulties (identifying medication; identifying medical supply; opening medication; opening medical supply) as distinct, binary events that occurred during the twelve months preceding the survey. Our survey did not afford respondents the opportunity to indicate how many times within the previous year a particular combination (identifying/opening by product type- medication or medical device had occurred). Option 2—where they report the problem in the binary, but participants were able to select all the different things that manifest this problem represents the Method most accurately. In retrospect, a collection of self-reported frequencies of occurrence would have added insight. That said, given that the goal of the work was to unearth whether (or not) packaging within the prehospital context was an area in need of further study for the purpose of validating the anecdotal evidence that we had, we feel like there is still value in this publication. Further, in light of the fact that the reported frequencies err on the side of conservatively estimating (under reporting the problems), we don’t see this as unduly problematic.

Nonetheless, we have added the following statement to our (newly added) Limitations section,

“A further limitation of the survey is the binary nature of the reporting of difficulties affiliated with packaging use. Specifically, a single question asks respondents about difficulties (identifying or opening) by product type (medications or medical device) and is only presented once. It asked respondents whether (or not) they had experienced a particular difficulty within the previous 12 months in binary fashion. A “yes response” triggered a cascade of questions that probed the difficulty and coping strategies that were employed as a result. Although the single question, binary nature of the frame provides us with some indication of the frequency across participants, the way that the survey was framed, we don’t have information about the number of occurrences for each of the respondents (within participant frequency). We also don’t have an indication of the number of times (across twelve months) a particular difficulty occurred and whether or not the coping mechanisms that they reported arose from multiple incidences or a single event.”

[REVIEWER #3]: The title of this paper need to change. Please restructure the title. Better remove ? on the title.

[AU]- The title has been modified. We suggest the following as the new title for the paper, which no longer includes the question that was previously present. 

“Paramedic interactions with the packaging of medications and medical supplies: Poor package design has the potential to impact patient health.”

[REVIEWER #3] Please make a proof ready on this paper.

Please do the proof read on this paper to make more technical and easy to understanding for the reader.

[AU] In addressing several of the comments, we believe we have enhanced the clarity of the paper. We have defined what is meant by “prehospital contexts” with the hope of improving clarity throughout, and attempted to reword where appropriate to enhance understanding across audiences.

6. PLOS authors have the option to publish the peer review history of their article (what does this mean?). If published, this will include your full peer review and any attached files.

Do you want your identity to be public for this peer review? For information about this choice, including consent withdrawal, please see our Privacy Policy.

Reviewer #1: No

Reviewer #2: No

Reviewer #3: No

---

## [Decision Letter · Decision Letter 1]

12 Jul 2021

Paramedic interactions with the packaging of medications and medical supplies: Poor package design has the potential to impact patient outcomes

PONE-D-20-33281R1

Dear Dr. Bix,

We’re pleased to inform you that your manuscript has been judged scientifically suitable for publication and will be formally accepted for publication once it meets all outstanding technical requirements.

Kind regards,

Dylan A Mordaunt

Academic Editor

PLOS ONE

Additional Editor Comments (optional):

Reviewers' comments:

Reviewer's Responses to Questions

**Comments to the Author**

1. If the authors have adequately addressed your comments raised in a previous round of review and you feel that this manuscript is now acceptable for publication, you may indicate that here to bypass the “Comments to the Author” section, enter your conflict of interest statement in the “Confidential to Editor” section, and submit your "Accept" recommendation.

Reviewer #2: All comments have been addressed

2. Is the manuscript technically sound, and do the data support the conclusions?

Reviewer #2: (No Response)

3. Has the statistical analysis been performed appropriately and rigorously? 

Reviewer #2: (No Response)

4. Have the authors made all data underlying the findings in their manuscript fully available?

Reviewer #2: (No Response)

5. Is the manuscript presented in an intelligible fashion and written in standard English?

Reviewer #2: (No Response)

6. Review Comments to the Author

Reviewer #2: I have no further no comments. All issues I raised in the first review have been addressed satisfactory.

7. PLOS authors have the option to publish the peer review history of their article (what does this mean?). If published, this will include your full peer review and any attached files.

Reviewer #2: No

---

## [Editor Report · Acceptance letter]

21 Jul 2021

PONE-D-20-33281R1 

Paramedic interactions with the packaging of medications and medical supplies: Poor package design has the potential to impact patient outcomes 

Dear Dr. Bix:

I'm pleased to inform you that your manuscript has been deemed suitable for publication in PLOS ONE. Congratulations! Your manuscript is now with our production department. 

Kind regards, 

on behalf of

Dr. Dylan A Mordaunt 

Academic Editor

PLOS ONE